# Characteristics and Outcomes of Patients with Invasive Pulmonary Aspergillosis and Respiratory Tract *Aspergillus* Colonization from a Tertiary University Hospital in Thailand

**DOI:** 10.3390/jof8040344

**Published:** 2022-03-25

**Authors:** Pannathat Soontrapa, Piriyaporn Chongtrakool, Methee Chayakulkeeree

**Affiliations:** 1Department of Medicine, Faculty of Medicine Siriraj Hospital, Mahidol University, Bangkok 10700, Thailand; pannathat2pong@gmail.com; 2Department of Microbiology, Faculty of Medicine Siriraj Hospital, Mahidol University, Bangkok 10700, Thailand; pchongtrakool@gmail.com; 3Division of Infectious Diseases and Tropical Medicine, Department of Medicine, Faculty of Medicine Siriraj Hospital, Mahidol University, Bangkok 10700, Thailand

**Keywords:** aspergillosis, *Aspergillus*, fungal infection, respiratory tract, mold

## Abstract

Positive culture for *Aspergillus* spp. from respiratory specimens needs to be interpreted together with relevant clinical conditions/settings to differentiate invasive infection from colonization. In this study, we aimed to investigate the association between positive culture for *Aspergillus* spp. from respiratory specimens and the presence of invasive pulmonary aspergillosis. Hospitalized patients with positive culture for *Aspergillus* spp. from any respiratory sample were retrospectively recruited. Patients were classified into two groups: those with invasive pulmonary aspergillosis and those with non-invasive aspergillosis/colonization. Two hundred and forty-one patients (48.1% male; mean age: 59.8 ± 14.5 years) were included. The most common *Aspergillus* spp. was *A. fumigatus* (21.0%). The most common underlying condition was chronic lung disease (23.7%), followed by solid tumor (22.4%). Myeloproliferative disease (aOR: 69.2, 95% CI: 2.4–1991.9), neutropenia ≥ 10 days (aOR: 31.8; 95% CI: 1.10–920.53), and corticosteroid treatment (aOR: 42.8, 95% CI: 6.5–281.3) were independent predictors of the invasive form. Chronic lung disease was independently inversely related to invasive form (OR: 0.04; 95% CI: 0.003–0.49). Serum galactomannan was positive in 69.2% of patients with invasive aspergillosis (OR: 25.9, 95% CI: 5.2–127.8). All inappropriately treated patients with invasive form died. In conclusion, positive culture for *Aspergillus* spp. from respiratory specimens with coexisting myeloproliferative disease, neutropenia ≥ 10 days, corticosteroid treatment, or positive serum galactomannan is highly suggestive of invasive pulmonary aspergillosis.

## 1. Introduction

Invasive pulmonary aspergillosis (IPA) is a critical medical condition with very high morbidity and mortality. Recent advancements in the care of critically ill patients have increased the incidence of IPA in both immunocompromised and immunocompetent individuals [1,2]. A diagnosis of IPA is made mainly based on the criteria proposed by the European Organization for Research and Treatment of Cancer/Mycosis Study Group (EORTC/MSG) [3], and these criteria have been effectively applied in immunocompromised hosts with classical risk factors. However, in patients with no classical risk factors, especially patients with critical illness with predisposing diseases other than neutropenia or organ transplantation, the diagnosis of IPA might be inconclusive and delayed. Suboptimal antifungal treatment in patients with IPA may cause disease progression and mortality [4,5,6].

With the finding of a positive culture for *Aspergillus* spp. from a respiratory tract specimen, suspicion of the invasive form of the disease is based on clinical and host factors [7,8]. Accurately distinguishing between *Aspergillus* colonization and invasive aspergillosis is a diagnostic challenge. Diagnosis of invasive aspergillosis in patients from whom *Aspergillus* was isolated from respiratory specimens should be interpreted together with other criteria, such as host factors, clinical symptoms and signs, and fungal markers [4,5,9]. The incidence of IPA in those with positive *Aspergillus* cultures from respiratory tract specimens was reported to be 20–40% [1,2]. Most previous studies focused on immunocompromised patients. Although the Asp-ICU algorithm has been proposed for diagnosis of invasive pulmonary aspergillosis in critically ill patients, there are currently no universal consensus criteria for the diagnosis of invasive aspergillosis in critically ill or immunocompetent individuals [5]. In fact, up to 1% of general intensive care unit (ICU) patients are found to be positive for *Aspergillus* spp. in the respiratory specimens, and both colonization and invasive disease are associated with poor outcomes [10]. Tissue diagnosis is usually required for definite diagnosis. Therefore, it is essentially necessary that we have the ability to identify patients at risk of invasive aspergillosis who do not satisfy the EORTC/MSG criteria [11]. Accordingly, the aims of this study were to compare the clinical features between invasive disease and colonization, to investigate the association between positive culture for *Aspergillus* spp. from respiratory specimens and the presence of invasive pulmonary aspergillosis, and to identify significant and independent risk factors for developing invasive pulmonary aspergillosis in hospitalized patients.

## 2. Materials and Methods

### 2.1. Study Design

This retrospective cohort study included hospitalized adult patients who had positive cultures for *Aspergillus* spp. from any respiratory tract samples. Adult patients aged more than 15 years old who were hospitalized at Siriraj Hospital, which is a tertiary care university hospital located in Bangkok, Thailand, during the period from January 2013 to March 2016 were enrolled. Eligible patients must have had one or more positive *Aspergillus* cultures from respiratory specimens, with or without risk factors and/or respiratory manifestations suspected of pulmonary aspergillosis. The respiratory tract specimens were sputa or bronchoscopic samples. This study was approved by the Siriraj Institutional Review Board (SIRB) (COA no. *Si* 366/2015) of the Faculty of Medicine Siriraj Hospital, Mahidol University. Written informed consent was not required for a retrospective study using the database from electronic medical records in our hospital.

Positive culture for *Aspergillus* spp. was defined as identification of *Aspergillus* colonies in at least one culture media tube. Clinical specimens were inoculated on Sabouraud dextrose agar (SDA), SDA containing chloramphenicol, and SDA containing chloramphenicol and cycloheximide and incubated at 25–27 °C for 14 days. All tube cultures were be observed three times a week. When there is mycelial growth, macroscopic appearance of the colony (both obverse and reverse), colony pigmentation, and topography will be noticed together with microscopic morphology of conidial structure and arrangements. During the study period, our microbiology laboratory routinely identified major pathogenic species of *Aspergillus*, i.e., *A. fumigatus, A. flavus, A. terreus*, and *A. niger* according to their macroscopic colony morphology, as well as microscopic conidial morphology and arrangement. *A.*
*fumigatus* colonies are blue-green and cottony on the obverse, consisting of a dense conidiophore with conical-shaped vesicle supporting a uniseriate phialide limited to the upper two-thirds. Conidia are produced in basipetal succession, forming long chains, and are globose to subglobose, green, and finely roughened. *A.*
*flavus* colonies are yellow-green in color, roughly granular, and often with radial grooves. Conidial heads are radiate with loose columns and biseriate with some phialides borne directly on the vesicle or uniseriate. Conidia are globose to subglobose, pale yellow-green, and echinulate. *A.*
*terreus* colonies are cinnamon buff to sand brown in color and powdery in appearance. Conidial heads are compact, columnar, and biseriate. Metulae are as long as the phialides. Conidiophore stipes are hyaline and smooth-walled. Conidia are globose to ellipsoidal, hyaline to slightly yellow, and smooth-walled. *A.*
*niger* colonies are black and granular, consisting of dense, dark-black conidial heads. Conidiophore stipes are smooth-walled and hyaline or turning dark towards the vesicle. Conidial heads are large, globose, and dark black, with radiate phialides. Conidial heads are biseriate with phialides borne on brown, often septate metulae. Conidia are globose to subglobose, dark brown to black, and rough-walled. Other *Aspergillus* spp. producing macroscopic colony color and morphology with microscopic conidial arrangements that were unable to match with these aforementioned characteristics were reported as *Aspergillus* species. Our hospital did not routinely perform molecular identification for all *Aspergillus* isolates. DNA sequencing and antifungal susceptibility tests were performed in some *Aspergillus* isolates according to physician requests, and therefore, these data were not completely included in this retrospective study. Antifungal susceptibility testing was performed by microbroth dilution method using Sensititre^®^ YeastOne^®^ plate (Thermo Fisher Scientific, Basingstoke, UK). Polymerase chain reaction (PCR) for molecular identification of *Aspergillus* DNA sequences was generated using primers to amplify the internal transcribed spacer region (*ITS1* and *ITS2*). The DNA sequences were subjected to an individual basic local alignment search tool (BLAST) in the GenBank database.

The criteria for invasive pulmonary aspergillosis and colonization/non-invasive aspergillosis were defined as follows:

Invasive pulmonary aspergillosis was defined as one of the following:Histopathological study compatible with IPA;Positive serum galactomannan index (>0.5) or bronchoalveolar lavage (BAL) galactomannan index (>1.0) and compatible imaging (halo, nodule, cavity, air crescent).

Colonization or non-invasive pulmonary aspergillosis was defined as the following:Pathological study compatible with other causes;No evidence of positive galactomannan and no compatible imaging;The patient survived for at least 30 days without antifungal treatment.

Inconclusive was defined as one of the following:No follow-up data;Positive SGM or BAL galactomannan but no compatible imaging.

### 2.2. Data Collection

Information gathered from electronic medical records included age, gender, clinical manifestation, onset of symptoms, associated diseases, predisposing immunosuppressive conditions, administration of broad-spectrum antibiotics, and use of corticosteroids. Laboratory data; chest roentgenogram; results of fungal culture and fungal marker; interventions, such as central line insertion, mechanical ventilation, and antifungal therapy; and treatment outcomes were recorded and analyzed.

Amphotericin B or voriconazole therapy was considered appropriate treatment for IPA. Isavuconazole was not available in our country during the study period. Patients with IPA who received other antifungal treatment, such as itraconazole, fluconazole, or any echinocandin, were considered to have received inappropriate treatment [12].

### 2.3. Statistical Analysis

Variables with a normally distributed continuous pattern are expressed as mean (SD), and differences between groups were assessed by Student’s *t*-test. Non-normally distributed continuous variables were expressed as median and quartiles, and differences between groups were assessed using Mann–Whitney U test. Discrete variables were compared with chi-square test or Fisher’s exact test depending on the size of the sample. Univariate and multivariate analyses were evaluated using binary logistic regression analysis (forward stepwise method) and presented as odds ratio (OR) (95% confidence interval (CI)). For all tests performed, a two-tailed *p*-value < 0.05 was considered to be statistically significant. PASW Statistic (SPSS) version 18.0 (SPSS, Inc., Chicago, IL, USA) was used to perform all statistical analyses.

## 3. Results

### 3.1. Microbiology

Regarding microbiologic findings, sputum cultures were performed in 62 cases, and the results revealed positive *Aspergillus* in 23 cases (37.1%). Positive tracheal suction cultures were found in 41 out of 55 cases (74.5%), positive BAL cultures in 175 out of 186 cases (94.1%), and positive *Aspergillus* from tissue culture in 12 out of 17 cases.

There were 8 cases of confirmed IPA by histopathological study from a total of 144 cases that underwent lung biopsy. Serum galactomannan was sent in seven cases, and three of those cases were positive.

A total of 266 *Aspergillus* respiratory isolates were identified from 241 patients enrolled in this study. The distribution of *Aspergillus* species isolated from respiratory specimens in patients with and without IPA is shown in Table 1. The most common *Aspergillus* species isolated from patients with IPA was *A. fumigatus* (55.0%), followed by *A. flavus* (40.0%). *A. niger* was the most common *Aspergillus* species to be isolated from those with colonization or non-IPA (48.3%). In our hospital, DNA sequencing and antifungal susceptibility testing for *Aspergillus* spp. were not routinely performed. Since 2015, we performed molecular identification and antifungal susceptibility tests in some *Aspergillus* spp. per physician requests. Only 24 isolates of *Aspergillus* spp. were requested for molecular identification. Among these 24 isolates, the most common species identified was *A. fumigatus* (12 isolates), followed by *A. flavus* (3 isolates); *A. terreus* (2 isolates); and one isolate each of *A. granulosus*, *A. gracilis/A. restrictus*, *A. penicillioides*, *A. syndowii*, and *A. tamarii*. The other two *Aspergillus* spp. isolates were unidentifiable by DNA sequencing. Only three *A. fumigatus* and one *A. terreus* isolates were requested for antifungal susceptibility testing. One *A. fumigatus* isolate and the *A. terreus* isolates were resistant to amphotericin B. All four isolates were susceptible to voriconazole, posaconazole, and itraconazole.

### 3.2. Patient Characteristics

During the study period, 254 patients met the inclusion criteria, and 4 patients were excluded due to insufficient data. Nine patients were equivocal for a diagnosis of IPA. Among the remaining 241 patients, the mean age was 59.8 ± 14.5 years, and 116 (48.1%) patients were male. There were 40 cases of IPA and 201 cases of *Aspergillus* colonization/non-invasive pulmonary aspergillosis (non-IPA). The mean age of patients with IPA was 52.9 ± 14.2 years, whereas that of non-IPA patients was 61.2 ± 14.2 years.

Patient characteristics compared between IPA and non-IPA by univariate analysis are shown in Table 2. Factors with a *p*-value < 0.2 in univariate analysis were entered into multivariate analysis to identify independent factors associated with IPA (Table 3). Multivariate analysis revealed myeloproliferative disease (adjusted odds ratio (aOR): 69.19, 95% confidence interval (Cl): 2.40–1,991.9; *p* = 0.01), prolonged neutropenia for at least 10 days (aOR: 31.76, 95%Cl: 1.10–920.53; *p* = 0.04), and corticosteroid treatment (aOR: 42.81, 95%Cl: 6.51–281.34; *p* < 0.001) to be independent predictors of IPA. Patient underlying diseases that required corticosteroid treatment comprised mainly autoimmune diseases (13/29, 44.8%) and hematologic malignancy (11/29, 37.9%). Interestingly, chronic lung disease was found to be inversely independently associated with IPA in our cohort (aOR: 0.04; 95% CI: 0.003–0.49; *p* = 0.01). In another analysis, serum galactomannan was found to be positive in 69.2% of patients with invasive aspergillosis (OR: 25.9, 95% CI: 5.2–127.8; *p* < 0.001).

Clinical manifestations and imaging compared between IPA and colonization/non-IPA are shown in Table 4 and Table 5, respectively. At least half of the parameters were found to be significantly different between groups in both analyses.

### 3.3. Diagnosis of IPA

From 241 patients, 40 were diagnosed as IPA in this study. When the EORTC/MSG criteria were applied, all 40 patients were diagnosed as either proven or probable IPA. When using the diagnostic criteria proposed by Blot et al. [5], 61 (25.3%) were classified as putative IPA. When comparing IPA diagnosed by EORTC/MSG and Blot et al. criteria, 34 persons were diagnosed as IPA and 174 as non-IPA. However, 27 critically ill patients diagnosed as IPA by Blot’s criteria were classified as non-IPA by EORTC/MSG criteria.

### 3.4. Outcomes

Among the 40 patients with IPA, 33 (82.5%) were treated with proper antifungal drugs (amphotericin B or voriconazole), and the other 7 patients received improper drugs. The 28-day mortality was 23/40 (57.5%). One patient with IPA proven by histopathological study had no follow-up data due to referral. Among those treated with proper antifungals, the mortality was 16/33 (48.5%). Of the 16 patients who died despite appropriate antifungal treatment, the cause of death in 12 patients was IPA. The causes of death in the other four patients were hospital-acquired pneumonia (two patients) and candidemia (two patients). In multivariate analysis, we found that factors associated with mortality in IPA patients who received appropriate antifungal treatment included lymphoproliferative disorder, endotracheal intubation, and having fever for more than 3 days (Table 6).

All six of the patients who received inappropriate treatment and for whom we had complete follow-up data died. Five of six patients died from IPA, and one patient died from hospital-acquired pneumonia.

Among the 201 patients without IPA, 17 (8.5%) patients died. The most common cause of death was hospital-acquired pneumonia (six patients), followed by tuberculosis (three patients), candidemia (two patients) and one each with mucormycosis, necrotizing fasciitis, pulmonary embolism, lung metastasis, liver abscess, and brain abscess. No patient succumbed from invasive aspergillosis in this group.

## 4. Discussion

Positive *Aspergillus* cultures from the respiratory tract do not reflect the presence of invasive disease in most patients. However, the coexistence of certain risk factors and positive *Aspergillus* cultures may increase the likelihood of IPA [13]. The present study investigated the relationship between positive culture for *Aspergillus* spp. from respiratory specimens and the presence of invasive pulmonary aspergillosis in Thailand, which is a resource-limited tropical country with a high prevalence of mold infections. The proportion of invasive pulmonary aspergillosis in this series was 16%, which is lower than the previously reported, ranging from 20% to 40% [2,5,6]. A recent Italian retrospective study in hospitalized patients with positive *Aspergillus* spp. culture from lower airway samples found that 66.6% of those fulfilled putative invasive pulmonary aspergillosis, which was much higher than our report [14]. However, the most common etiologic agent of invasive disease was still *A. fumigatus*, which is the most important pathogenic *Aspergillus* species. Although molecular identification was not performed in all *Aspergillus* isolates, our limited data still showed that most of the isolates identified from DNA sequencing were *A. fumigatus*, with a number of emerging cryptic species identified.

The independent risk factors for IPA that were identified in our study included the presence of myeloproliferative disease, prolonged neutropenia (defined as neutropenia lasting longer than 10 days), and corticosteroid treatment. Consistent with our findings, a previous study reported that 50% of IPA cases had hematologic malignancy, 64% had neutropenia, and 20% had corticosteroid treatment [13]. In contrast to previous studies, we found no association between critically ill condition and IPA. A study conducted in Belgium found that 60% (50/83) of ICU patients with IPA had no classical risk factors for IPA, such as neutropenia, hematologic malignancy, or stem cell or bone marrow transplantation [2,15].

A study from China reported that chronic obstructive pulmonary disease (COPD) patients with positive *Aspergillus* spp. cultures from lower respiratory tract samples had similar IPA incidence, mortality, and survival time as immunocompromised individuals. Therefore, they reported COPD to be a potential risk factor for IPA, especially in patients admitted to the ICU [9]. In interesting contrast, our study found COPD to have a negative independent correlation to IPA, although some of those patients had corticosteroid treatment.

Recently reported emerging non-classical risk factors for IPA include critical illness, chronic pulmonary disease, biologic agents, small-molecule kinase inhibitors, and viral pneumonia [4]. In Thailand, classic risk factors, such as those mentioned above, still play a significant role in the pathogenesis of IPA. This may be partly due to a relatively lower use of immunosuppressive agents in our country. Nevertheless, there is a trend toward a change in risk factors from classical to non-classical due to advances in medical care.

Unlike a study conducted in an acute respiratory distress syndrome (ARDS) setting [16] in which positive BAL galactomannan but not serum galactomannan was well correlated with IPA, the present study found serum galactomannan to be positive in the majority of patients diagnosed with IPA. Blot’s criteria for putative aspergillosis in critically ill patients [5] include characteristics that effectively discriminate IPA from colonization. With that in mind, our study found the EORTC/MSG criteria to be more appropriate in our setting; however, only 12.9% of our overall study population were admitted to the ICU.

The clinical outcome of invasive pulmonary aspergillosis depends on prompt initiation of appropriate antifungal treatment. Voriconazole is the recommended first-line therapy, whereas amphotericin B is optional and should be reserved for use only in certain settings [4,11,12]. Our study found 100% mortality among patients that received inappropriate treatment. Despite receiving appropriate antifungal treatment, approximately half of patients with IPA died, mostly from IPA itself. We found that lymphoproliferative disorder, endotracheal intubation, and fever for more than 3 days were the independent risk factors for mortality. These findings imply an association between survival and disease severity and the underlying immunocompromised condition of patients with IPA. In contrast, of 17 non-IPA patients who died, none died from IPA. These results indicate that although *Aspergillus* spp. was isolated from the respiratory tract in non-IPA patients, the fungus is not likely to develop invasive disease and cause death.

The main limitation of this study is its single-center retrospective design, which potentially limits of the generalizability of our findings and renders it vulnerable to missing or incomplete data. In addition, according to the retrospective nature of this study, we did not have information regarding the molecular identification of all isolates, and therefore, future studies are warranted to investigate the cryptic species of pathogenic *Aspergillus* spp. A prospective cohort fungal registry is ongoing in our hospital.

## 5. Conclusions

Positive culture for *Aspergillus* spp. from respiratory specimens with coexisting myeloproliferative disease, neutropenia ≥ 10 days, corticosteroid treatment, or positive serum galactomannan is highly suggestive of invasive pulmonary aspergillosis. Timely administration of appropriate antifungal drugs is crucial for reducing mortality.

## Figures and Tables

**Table 1 jof-08-00344-t001:** *Aspergillus* species identified from respiratory specimen cultures in patients with and without invasive pulmonary aspergillosis.

Aspergillus Species	IPA (*n* = 40), *n* (%)	Colonization/Non-IPA (*n* = 201), *n* (%)	*p*-Value
*Aspergillus fumigatus*	22 (55.0)	34 (16.9)	<0.001
*Aspergillus flavus*	16 (40.0)	35 (17.4)	0.001
*Aspergillus niger*	4 (10.0)	97 (48.3)	<0.001
*Aspergillus terreus*	1(2.5)	1 (0.5)	0.305
*Aspergillus* spp.	6 (15.0)	50 (24.9)	0.177

Abbreviations: IPA: invasive pulmonary aspergillosis.

**Table 2 jof-08-00344-t002:** Patient demographic and clinical characteristics.

Parameters	All Patients (*n* = 241)	IPA (*n* = 40)	Colonization/ Non-IPA (*n* = 201)	*p*-Value	Odds Ratio (95% CI)
Age (years, mean ± SD)	59.8 ± 14.5	52.9 ± 14.2	61.2 ± 14.2	0.001	0.96 (0.94–0.99)
Male	116 (48.1)	22 (55.0)	94 (46.8)	0.34	1.39 (0.70–2.75)
Diabetes mellitus	40 (16.6)	6 (15.0)	34 (16.9)	0.77	0.87 (0.34–0.23)
CKD	46 (19.1)	12 (30.0)	34 (16.9)	0.054	2.10 (0.97–4.55)
Chronic lung disease	57 (23.7)	1 (2.5)	56 (27.9)	0.001	0.07 (0.01–0.50)
Previous pulmonary tuberculosis	34 (14.1)	3 (7.5)	31 (15.4)	0.19	0.45 (0.13–1.53)
Chronic liver disease	13 (5.4)	4 (10.0)	9 (4.5)	0.24	2.37 (0.69–8.11)
Myeloproliferative disease	9 (3.7)	8 (20.0)	1 (0.5)	<0.001	50.00 (6.05–413.27)
Lymphoproliferative disease	15 (6.2)	9 (22.5)	6 (3.0)	<0.001	9.44 (3.14–28.36)
Solid tumor	54 (22.4)	1 (2.5)	53 (26.4)	0.001	0.07 (0.01–0.53)
Solid organ transplant	3 (1.2)	2 (5.0)	1 (0.5)	0.07	10.53 (0.93–119.02)
Connective tissue disease	28 (11.6)	13 (32.5)	15 (7.5)	<0.001	5.97 (2.56–13.90)
Neutropenia	20 (8.3)	16 (40.0)	4 (2.0)	<0.001	5.97 (2.56–13.90)
Prolonged neutropenia ≥ 10 days	9 (3.7)	8 (20.0)	1 (0.5)	<0.001	50.00 (6.05–413.27)
T-cell immunosuppressive drugs *	46 (19.1)	27 (67.5)	19 (9.5)	<0.001	19.90 (8.82–44.86)
Corticosteroid **	53 (22.0)	29 (72.5)	24 (11.9)	<0.001	19.44 (8.61–43.91)
Steroid duration, median (IQR)	18 (10–60)	47 (16–71)	13 (5–40)	0.009	0.999 (0.996–1.002)
Previous antifungal treatment within 3 months	17 (7.1)	8 (20.0)	9 (4.5)	0.002	5.33 (1.92–14.84)
Previous antibiotics treatment within 3 months	116 (48.1)	36 (90.0)	80 (39.8)	<0.001	13.61 (4.67–39.72)
Endotracheal intubation	46 (19.1)	19 (47.5)	27 (13.4)	<0.001	5.83 (2.78–12.24)
ICU admission	31 (12.9)	11 (27.5)	20 (10.0)	0.002	3.43 (1.49–7.90)
RRT	16 (6.6)	7 (17.5)	9 (4.5)	0.01	4.53 (1.58–12.99)
Vasopressor	18 (7.5)	4 (10.0)	14 (7.0)	0.51	1.48 (0.46–4.77)
Central line insertion	30 (12.4)	10 (25.0)	20 (10.0)	0.02	3.02 (1.29–7.07)

Unless otherwise indicated, all data are shown in *n* (%). Abbreviations: CKD: chronic kidney disease; ICU: intensive care unit; RRT: renal replacement therapy; IPA: invasive pulmonary aspergillosis. * Previous t-cell immunosuppressive drug treatment within 3 months. ** Previous steroid treatment with at least 300 mg of prednisolone or equivalent dose within 3 months.

**Table 3 jof-08-00344-t003:** Multivariate analysis for factors independently associated with the development of invasive aspergillosis.

Parameters	All Patients (*n* = 241)	IPA (*n* = 40)	Colonization/ Non-IPA (*n* = 201)	*p*-Value	Adjusted Odds Ratio (95% CI)
Age (years, mean ± SD)	59.8 ± 14.5	52.9 ± 14.2	61.2 ± 14.2	0.27	0.98 (0.94–1.02)
Chronic lung disease	57 (23.7)	1 (2.5)	56 (27.9)	0.01	0.04 (0.003–0.49)
CKD	46 (19.1)	12 (30.0)	34 (16.9)	0.34	0.44 (0.08–2.38)
Previous pulmonary tuberculosis	34 (14.1)	3 (7.5)	31 (15.4)	0.76	1.39 (0.18–10.80)
Myeloproliferative disease	9 (3.7)	8 (20.0)	1 (0.5)	0.01	69.19 (2.40–1991.90)
Lymphoproliferative disease	15 (6.2)	9 (22.5)	6 (3.0)	0.80	0.77 (0.10–5.89)
Connective tissue disease	28 (11.6)	13 (32.5)	15 (7.5)	0.16	3.14 (0.64–15.47)
Neutropenia	20 (8.3)	16 (40.0)	4 (2.0)	0.21	3.68 (0.48–28.54)
Prolonged neutropenia ≥ 10 days	9 (3.7)	8 (20.0)	1 (0.5)	0.04	31.76 (1.10–920.53)
T-cell immunosuppressive drugs *	46 (19.1)	27 (67.5)	19 (9.5)	0.48	1.65 (0.41–6.69)
Corticosteroid **	53 (22.0)	29 (72.5)	24 (11.9)	<0.001	42.81 (6.51–281.34)
Previous antifungal treatment within 3 months	17 (7.1)	8 (20.0)	9 (4.5)	0.34	0.41 (0.07–2.55)
Previous antibiotics treatment within 3 months	116 (48.1)	36 (90.0)	80 (39.8)	0.57	1.64 (0.30–9.04)
Endotracheal intubation	46 (19.1)	19 (47.5)	27 (13.4)	0.41	2.08 (0.36–11.94)
ICU admission	31 (12.9)	11 (27.5)	20 (10.0)	0.49	1.89 (0.31–11.67)
RRT	16 (6.6)	7 (17.5)	9 (4.5)	0.93	0.89 (0.71–11.27)
Central line insertion	30 (12.4)	10 (25.0)	20 (10.0)	0.80	0.75 (0.08–7.32)

Unless otherwise indicated, all data are shown in *n* (%). Abbreviations: CKD: chronic kidney disease; ICU: intensive care unit; RRT: renal replacement therapy; IPA: invasive pulmonary aspergillosis * Previous t-cell immunosuppressive drug treatment within 3 months ** Previous steroid treatment with at least 300 mg of prednisolone or equivalent dose within 3 months.

**Table 4 jof-08-00344-t004:** Clinical manifestations and galactomannan level compared between patients with invasive pulmonary aspergillosis (IPA) and patients with colonization or non-IPA.

Symptoms	IPA (*n* = 40)	Colonization/ Non-IPA (*n* = 201)	*p*-Value	Odds Ratio (95% CI)
Fever (body temperature ≥ 37.8 °C)	24 (60.0)	16 (8.0)	<0.001	17.34 (7.69–39.11)
Pleuritic chest pain	3 (7.5)	16 (8.0)	1.00	0.94 (0.26–3.38)
Dyspnea	33 (82.5)	90 (44.8)	<0.001	5.81 (2.46–13.76)
Hemoptysis	8 (20.0)	34 (16.9)	0.64	1.23 (0.52–2.90)
Positive SGM	27/39 (69.2)	2/25 (8.0)	<0.001	25.88 (5.24–127.77)
Positive BAL GM	9/12 (75.0)	0/1 (0)	0.31	-

Abbreviations: BAL GM: bronchoalveolar lavage galactomannan; SGM: serum galactomannan. Optical density (OD) ratio of 0.5 or greater for serum galactomannan and 1.0 for BAL fluid was considered positive.

**Table 5 jof-08-00344-t005:** Chest computed tomography (CT) or chest radiography compared between patients with invasive pulmonary aspergillosis (IPA) and patients with colonization or non-IPA.

Imaging	IPA (*n* = 40)	Colonization/ Non-IPA (*n* = 201)	*p*-Value	Odds Ratio (95% CI)
Diffuse reticular or alveolar opacities	23 (57.5)	77 (38.3)	0.02	2.18 (1.10–4.34))
Non-specific infiltration/consolidation	26 (65.0)	108 (53.7)	0.19	1.60 (0.79–3.24)
Pleural effusion	18 (45.0)	41 (20.4)	0.001	3.19 (1.57–6.50)
Wedge-shaped infiltration	4 (10.0)	2 (1.0)	0.008	11.06 (1.95–62.62)
Well-shaped nodule *	29 (72.5)	81 (40.3)	<0.001	3.91 (1.85–8.26)
Air crescent sign	1 (2.5)	1 (0.5)	0.31	5.13 (0.31–83.74)
Halo sign	16 (40.0)	2 (1.0)	<0.001	66.33 (14.37–306.27)
Cavitation	13 (32.5)	25 (12.4)	0.001	3.39 (1.55–7.42)
Typical (air crescent/halo/cavitation/nodule)	34 (85.0)	94 (46.8)	<0.001	6.45 (2.59–16.04)

* Pulmonary nodules include mass at a size of 5 mm to 3 cm in diameter.

**Table 6 jof-08-00344-t006:** Univariate and multivariate analyses for factors significantly associated with mortality in patients with IPA.

	Univariate Analysis				Multivariate Analysis	
Risk Factors	Survived (*n* = 17), *n* (%)	Died (*n* = 16), *n* (%)	Odds Ratio (95% CI)	*p*-Value	Adjusted Odds Ratio (95% CI)	*p*-Value
Lymphoproliferative disorder	1 (5.9%)	6 (37.5%)	9.6 (1.002–91.96)	0.049	41.46 (1.83–941.35)	0.019
T-cell immunosuppressive therapy	10 (58.8%)	15 (93.8%)	10.50 (1.12–98.91)	0.040	-	-
Endotracheal intubation	3 (17.6%)	10 (62.5%)	7.78 (1.56–38.76)	0.012	45.38 (2.98–691.91)	0.006
Fever > 3 days	8 (47.1%)	13 (81.2%)	4.88 (1.01–23.57)	0.049	18.37 (1.28–263.32)	0.032
Diffused lung opacities	6 (35.3%)	12 (75.0%)	5.50 (1.22–24.81)	0.027	-	-

A *p*-value < 0.05 indicates statistical significance. Abbreviation: CI: confidence interval; IPA: invasive pulmonary aspergillosis.

## Data Availability

No new data were created or analyzed in this study. Data sharing is not applicable to this study.

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
