# Peer review of "Characteristics and Outcomes of Patients with Invasive Pulmonary Aspergillosis and Respiratory Tract Aspergillus Colonization from a Tertiary University Hospital in Thailand"

_jof, 2022, doi:10.3390/jof8040344_

Round 1
Reviewer 1 Report
The article submitted by Soontrapa and co-authors is a retrospective study of the clinical characteristic of 250 patients with invasive pulmonary aspergillosis from a tertiary university hospital in Thailand. By using appropriate statistical analysis, they compared invasive pulmonary aspergillosis vs non- invasive pulmonary aspergillosis/colonization. This is a large dataset in a single hospital so as to draw general assumptions, as the authors mention in the discussion section. They concluded that positive culture for Aspergillus spp. from respiratory specimens with coexisting myeloproliferative disease, neutropenia ≥10 days, corticosteroid treatment, or positive serum galactomannan is highly suggestive of invasive pulmonary aspergillosis. Timely administration of appropriate antifungal drugs is crucial for reducing mortality. It is a comprehensive, well- organized and presented study. Minor revisions are necessary, so as to improve the manuscript. In particular,
Page 1:
The title is too long. A suggestive title is: Characteristics and outcomes of patients with invasine pulmonary aspergillosis from a tertiary university hospital in Thailand.
Page 2:
Please replace ''Patients were classified as'' with ''Patients were classified into two groups: those with invasive pulmonary aspergillosis or non-invasive aspergillosis/colonization.
Page 3:
- Please replace ''Patients were classified as'' with ''Patients were classified into two groups: those with invasive pulmonary aspergillosis or non-invasive aspergillosis/colonization.
- Please replace catastrophic with a synonymous word, such as critical
- The authors are advised to refer to the updated criteria for Invasive aspergillosis, which have been published in 2020 and correct any changes in the text and categorization of the patients’ data for analysis (Donnelly, J.P.; Chen, S.C.; Kauffman, C.A.; Steinbach, W.J.; Baddley, J.W.; Verweij, P.E.; Clancy, C.J.; Wingard, J.R.; Lockhart, S.R.; Groll, A.H.; et al. Revision and Update of the Consensus Definitions of Invasive Fungal Disease From the European Organization for Research and Treatment of Cancer and the Mycoses Study Group Education and Research Consortium.  Infect. Dis. 2020, 71, 1367–1376. doi:HYPERLINK "https://doi.org/10.1093/CID/CIZ1008"10.1093/CID/CIZ1008.)
- Please clarify the sentence. ‘’There are currently no consensus criteria for the diagnosis of invasive aspergillosis in critically ill or immunocompetent individuals.’’As the authors cited in ref. 5 (Blot et al. 2012), the Asp-ICU algorithm has been proposed for discriminate Aspergillus respiratory tract colonization from invasive pulmonary aspergillosis in critically ill patients.
Page 7:
- Please describe the microbiological methods (media, time and temperatures of growth, microscopic examination) in the Materials and Methods section.
Page 8:
- Please describe the method used for antifungal susceptibility testing and molecular identification in the Material and Methods section.
Page 12:
- The authors mention that ''However, 27 patients diagnosed as IPA by Blot’s criteria were classified as non-IPA by EORTC/MSG criteria'' Blot's criteria have been suggested for critically ill patients. Please clarify whether any of these 27 patients were critically-ill or transferred in an ICU?
Page 13& 15:
Which are the ''cryptic'' species?
Pages 17-19:
Assure consistency in capitalization usage in titles of articles references.
Author Response
Dear Reviewer 1
Thank you very much for your constructive comments. Please see our response to your comments and the revised manuscript submitted. We hope that the revised version would improve the quality of the manuscript. The revised texts in the manuscript are now highlighted in YELLOW in the revised manuscript.
Comments from reviewer:
The article submitted by Soontrapa and co-authors is a retrospective study of the clinical characteristic of 250 patients with invasive pulmonary aspergillosis from a tertiary university hospital in Thailand. By using appropriate statistical analysis, they compared invasive pulmonary aspergillosis vs non- invasive pulmonary aspergillosis/colonization. This is a large dataset in a single hospital so as to draw general assumptions, as the authors mention in the discussion section. They concluded that positive culture for Aspergillus spp. from respiratory specimens with coexisting myeloproliferative disease, neutropenia ≥10 days, corticosteroid treatment, or positive serum galactomannan is highly suggestive of invasive pulmonary aspergillosis. Timely administration of appropriate antifungal drugs is crucial for reducing mortality. It is a comprehensive, well- organized and presented study. Minor revisions are necessary, so as to improve the manuscript. In particular,
Response: Thank you for your positive comments.
Page 1:
The title is too long. A suggestive title is: Characteristics and outcomes of patients with invasive pulmonary aspergillosis from a tertiary university hospital in Thailand.
Response: Thank you for your suggestion. This manuscript addressed not only the characteristics and outcomes of invasive pulmonary aspergillosis, but also Aspergillus colonization. Therefore, we have shortened the title of the manuscript as suggested by the reviewer with modification.
The title of the manuscript has been changed to “Characteristics and Outcomes of Patients with Invasive Pulmonary Aspergillosis and Respiratory Tract Aspergillus Colonization from A Tertiary University Hospital in Thailand.”
Page 2:
Please replace ''Patients were classified as'' with ''Patients were classified into two groups: those with invasive pulmonary aspergillosis or non-invasive aspergillosis/colonization.
Response: We have replaced the phrase in the Abstract as suggested by the reviewer.
Page 3:
Please replace catastrophic with a synonymous word, such as critical
Response: We have replaced the word (“catastrophic” to “critical”) in the Introduction section as suggested by the reviewer.
The authors are advised to refer to the updated criteria for Invasive aspergillosis, which have been published in 2020 and correct any changes in the text and categorization of the patients’ data for analysis (Donnelly, J.P.; Chen, S.C.; Kauffman, C.A.; Steinbach, W.J.; Baddley, J.W.; Verweij, P.E.; Clancy, C.J.; Wingard, J.R.; Lockhart, S.R.; Groll, A.H.; et al. Revision and Update of the Consensus Definitions of Invasive Fungal Disease From the European Organization for Research and Treatment of Cancer and the Mycoses Study Group Education and Research Consortium.  Infect. Dis. 2020, 71, 1367–1376. doi:HYPERLINK "https://doi.org/10.1093/CID/CIZ1008"10.1093/CID/CIZ1008.)
Response: Thank you for the suggestion. The updated EORTC/MSG definition 2020 was not released during the study period. However, we have applied the 2020 EORTC/MSG criteria to our work and we found that the patient categorization and results were not changed. We therefore revised the reference from the 2008 version to the 2020 version as suggested by the reviewer.
Please clarify the sentence. “There are currently no consensus criteria for the diagnosis of invasive aspergillosis in critically ill or immunocompetent individuals.” As the authors cited in ref. 5 (Blot et al. 2012), the Asp-ICU algorithm has been proposed for discriminate Aspergillus respiratory tract colonization from invasive pulmonary aspergillosis in critically ill patients.
Response: We agree that the Asp-ICU algorithm is a well-known clinical algorithm that has been proposed the criteria for diagnosis of invasive pulmonary aspergillosis in critically ill patients. However, it has not been widely used in clinical settings nor adopted in international guidelines.
We therefore revised the sentence “There are currently no consensus criteria for the diagnosis of invasive aspergillosis in critically ill or immunocompetent individuals” to “Although the Asp-ICU algorithm has been proposed for diagnosis of invasive pulmonary aspergillosis in critically ill patients, there are currently no universal consensus criteria for the diagnosis of invasive aspergillosis in critically ill or immunocompetent individuals [5].” (page 3 with a reference added)
Page 7:
Please describe the microbiological methods (media, time and temperatures of growth, microscopic examination) in the Materials and Methods section.
Response: The following sentences were added to Materials and Methods section as suggested by the reviewer.
“Clinical specimens were inoculated on Sabouraud Dextrose Agar (SDA), SDA containing chloramphenicol, SDA containing chloramphenicol and cycloheximide and incubated at 25-27°C for 14 days. All tube cultures will be observed three times a week. When there is mycelial growth, macroscopic appearance of the colony, both obverse and reverse, colony pigmentation as well as topography will be noticed together with microscopic morphology of conidial structure and arrangements. (page 4-5)
Page 8:
Please describe the method used for antifungal susceptibility testing and molecular identification in the Material and Methods section.
Response: The following sentences were added to Materials and Methods section as suggested by the reviewer.
“Antifungal susceptibility testing was performed by microbroth dilution method using Sensititre® Yeast One® plate (Thermo Fisher Scientific, Basingstoke, UK). Polymerase chain reaction (PCR) for molecular identification of Aspergillus DNA sequences was generated using primers to amplify internal transcribed spacer region (ITS1 and ITS2). The DNA sequences were subjected to an individual Basic Local Alignment Search Tool (BLAST) in GenBank database.” (page 6)
Page 12:
The authors mention that ''However, 27 patients diagnosed as IPA by Blot’s criteria were classified as non-IPA by EORTC/MSG criteria'' Blot's criteria have been suggested for critically ill patients. Please clarify whether any of these 27 patients were critically-ill or transferred in an ICU?
Response: Thank you for your comments. All of the 27 patients were in critically ill condition. Therefore, we revised the sentence as “However, 27 critically ill patients diagnosed as IPA by Blot’s criteria were classified as non-IPA by EORTC/MSG criteria.” (page 10)
Page 13& 15:
Which are the ''cryptic'' species?
Response: The Cryptic species are the obscure species that are difficult or not able to identify by conventional methods. We have mentioned in the “Microbiology” section under the “Results” section (page 8) about the Cryptic species in our study as below;
“Only 24 isolates of Aspergillus spp. were requested for molecular identification. Among these 24 isolates, the most common species identified was A. fumigatus (12 isolates), followed by A. flavus (3 isolates), A. terreus (2 isolates), and one isolate each of A. granulosus, A. gracilis/A. restrictus, A. penicillioides, A. syndowii, and A. tamarii. The other 2 Aspergillus spp. isolates were unidentifiable by DNA sequencing.”
Pages 17-19:
Assure consistency in capitalization usage in titles of articles references.
Response: We have corrected as suggested by the reviewer.
Regards,
Methee Chayakulkeeree, MD, PhD, FECMM
Corresponding author on behalf of all authors

Reviewer 2 Report
The authors presented a paper about invasive pulmonary aspergillosis by specifically describing characteristics and outcomes. They focused on samples derived from the respiratory tract of patients, studying the opportunity to improve diagnosis and to predict significant and independent risk factors for developing invasive pulmonary aspergillosis in hospitalized patients.
The paper is overall well-written and detailed. Many informative tables are able to describe and give information about patients characteristics and statistical analyses to understand possible correlations among several factors: this is a strength of the paper, in my opinion.
I would suggest to consider the possibility to move the description of the single Aspergillus species characteristics from "study design" to "microbiology" part in the results section: this is only a suggestion, not a mandatory request.
Moreover, it would be useful to explain how the authors consider Amphotericin B or voriconazole as appropriate antifungal treatments while itraconazole, fluconazole, or any echinocandin, were considered as inappropriate treatment, possibly justifying with references.
This paper lacks a good references section in supporting both the study and the field of interest. This is a scientific field very well studied and discussed, then I think that the author must improve this part, including many other reviews/articles that are present in the past and current literature.
The sentence "Written informed consent was not obtained due to the confidential retrospective nature of this study" I think is not completely acceptable, because also in retrospective studies is mandatory to have an informed consent, in order to collect and store the samples for a specific time, but this probably depends on specific countries rules. Can the authors clarify this point?
Author Response
Dear Reviewer 2
Thank you very much for your constructive comments. Please see our response to your comments and the revised manuscript submitted. We hope that the revised version would improve the quality of the manuscript. The revised texts in the manuscript are now highlighted in YELLOW in the revised manuscript.
Comments from reviewer:
The authors presented a paper about invasive pulmonary aspergillosis by specifically describing characteristics and outcomes. They focused on samples derived from the respiratory tract of patients, studying the opportunity to improve diagnosis and to predict significant and independent risk factors for developing invasive pulmonary aspergillosis in hospitalized patients.
The paper is overall well-written and detailed. Many informative tables are able to describe and give information about patients characteristics and statistical analyses to understand possible correlations among several factors: this is a strength of the paper, in my opinion.
I would suggest to consider the possibility to move the description of the single Aspergillus species characteristics from "study design" to "microbiology" part in the results section: this is only a suggestion, not a mandatory request.
Response: Than you for your suggestion. We also received a comment from another reviewer to add some detail related to Aspergillus identification in the Materials and Methods section. According to your suggestion, we have tried to reshuffle the text describing single Aspergillus species characteristics from materials and Methods section to Microbiology section. However, we feel that the flow of the presentation in the manuscript is better and easier to follow as it is in the original place. Therefore, we decided to leave this part in the same place without revision.
Moreover, it would be useful to explain how the authors consider Amphotericin B or voriconazole as appropriate antifungal treatments while itraconazole, fluconazole, or any echinocandin, were considered as inappropriate treatment, possibly justifying with references.
Response: According to the Practice Guidelines for the Diagnosis and Management of Aspergillosis: 2016 Update by the Infectious Diseases Society of America (Patterson TF, et al. Clinical Infectious Diseases 2016;63(4):e1–60), antifungal treatment for IPA was suggested as below;
- Voriconazole and isavuconazole are the first line treatment for IPA.
- Amphotericin B deoxycholate and its lipid derivatives are appropriate options for initial and salvage therapy of Aspergillus infections when voriconazole cannot be administered.
- Itraconazole is rarely recommended in patients with acute IPA, with its use reserved for patients with less severe or less invasive disease presentations.
- Fluconazole does not have activity against Aspergillus
- Echinocandins are not recommend their routine use as monotherapy for the primary treatment of IA
Therefore, we consider Amphotericin B or voriconazole as appropriate antifungal treatments (isavuconazole was not available during the study time). We have added the reference above in the text as “Patients with IPA who received other antifungal treatment such as itraconazole, fluconazole, or any echinocandin, were considered as those with inappropriate treatment [12].” (page 7)
This paper lacks a good references section in supporting both the study and the field of interest. This is a scientific field very well studied and discussed, then I think that the author must improve this part, including many other reviews/articles that are present in the past and current literature.
Response: Thank you very much for your advice. We added 4 more recently published references to out manuscript. The reference is now 16. We also added/modified texts that were relevant to the references as follow;
- Page 3, reference 7,8 - With the finding of a positive culture for Aspergillus spp. from a respiratory tract specimen, suspicion of the invasive form of the disease is based on clinical and host factors [7, 8].
- Page 3-4, reference 10 - In fact, up to 1% of general intensive care unit (ICU) patients reveal positive Aspergillus spp. in the respiratory specimens and both colonization and invasive disease are associated with poor outcomes [10].
- Page 14, reference 11- A recent Italian retrospective study in hospitalized patients with positive Aspergillus spp. culture from lower airway samples and they found that 66.6% of those fulfilled putative invasive pulmonary aspergillosis, which was much higher than our report [14].
The sentence "Written informed consent was not obtained due to the confidential retrospective nature of this study" I think is not completely acceptable, because also in retrospective studies is mandatory to have an informed consent, in order to collect and store the samples for a specific time, but this probably depends on specific countries rules. Can the authors clarify this point?
Response: This is a retrospective study using the data from electronic medical record without collecting or storing any samples. The data are from the real-life medical practice. We modified the sentence as “Written informed consent was not required for a retrospective study using the database from electronic medical records in our hospital.” (page 4)
Regards,
Methee Chayakulkeeree, MD, PhD, FECMM
Corresponding author on behalf of all authors
